# Effect of Menstrual Cycle Phase on Fuel Oxidation Post HIT in Women Reproductive Age: A Pilot Study

**DOI:** 10.3390/ijerph20043148

**Published:** 2023-02-10

**Authors:** Caroline Santana Frientes, Marcelo Luis Marquezi, Juliana Monique Lino Aparecido, Marcelo Santin Cascapera, Patrícia Soares Rogeri, Antônio Herbert Lancha Junior

**Affiliations:** 1Laboratory of Physical Education Research (LAPEF), University City of Sao Paulo (UNICID), Sao Paulo 05508-030, Brazil; 2Pediatric Cardiology Group, Departament of Pediatrics and Childcare of Irmandade da Santa Casa de Misericórdia de São Paulo (ISCMSP), Sao Paulo 01221-010, Brazil; 3Laboratory of Clinical Investigation, Experimental Surgery (LIM 26), Clinic’s Hospital of Medical School, University of Sao Paulo, Sao Paulo 05508-030, Brazil

**Keywords:** women, exercise, energy metabolism, menstrual cycle, high-intensity interval training

## Abstract

Women of childbearing age have variations in substrate oxidation rates that can lead to overweight, type II diabetes, and other conditions that may be associated with metabolic inflexibility and the variations in estrogen concentrations observed during the monthly ovarian cycle. Purpose: This study aimed to verify and compare the influence of eight treadmill high-intensity interval training (HIT) sessions on carbohydrate and lipid oxidation rates (CHOox and LIPox, respectively) and intensities of ventilatory anaerobic thresholds (VATs) of women in different phases of the monthly ovarian cycle. Methods: Eleven irregularly active women performed incremental treadmill exercise testing followed by submaximal work-rate running for 45 min to determine VATs, VO_2peak_, peak velocity (V_peak_), and substrate oxidation rates, before and after a training period, in different phases of their monthly ovarian cycle (follicular phase group, FL, *n* = 6; luteal phase group, LT, *n* = 5). The training period consisted of eight HIT sessions, composed each one of eight sets of 60 s running at 100%V_peak_ interspersed by 75 s recovery every 48 h. Results: Our results showed no significant differences in VATs intensities between groups. The comparison between groups showed significant differences in relative energy derived from CHO pre- and post-training of −61.42% and −59.26%, respectively, and LIP pre- and post-training of 27.46% and 34.41%, respectively. The relative energy derived from CHO after the training period was 18.89% and 25.50% higher for FL and LT, respectively; consequently, the relative energy derived from LIPox after the training period was 8,45% and 3.46% lower for FL and LT, respectively. Over the training period, V_peak_ was ~13.5 km/h, which produced the relative intensities of ~89%VO_2peak_ e ~93%HR_peak_ for both groups. Conclusion: The monthly ovarian cycle phases promote significant changes in substrate oxidation rates leading to a decrease in CHOox. High-intensity interval training can minimize the differences observed and constitute an alternative intervention.

## 1. Introduction

Women of childbearing age, apparently healthy, may have a high propensity for changes in the oxidation of energy substrates (lipids and carbohydrates) and insulin resistance development, which, if not detected, can progress to type II diabetes, obesity, or other associated pathologies. This insulin resistance can occur due to metabolic inflexibility, the body’s inability to use energy substrates as fuel, and the transition between them in response to dietary change, energy availability, or circulating substrate concentrations [1,2,3].

During the fertile period, women experience a monthly rhythm called the menstrual cycle. Each cycle lasts an average of 23 to 28 days and can be divided into follicular and luteal phases, in which hormonal fluctuations (estrogen and progesterone) occur. These hormones regulate reproductive function and other physiological systems, such as the respiratory, thermoregulatory, and cardiovascular systems, which can affect exercise performance. Most studies carried out with women prefer the follicular phase and close to ovulation, in which estrogen has its highest concentration. During this phase, more remarkable changes are noted in the oxidation rates of energy substrates, including increased availability of fatty acids, oxidative capacity, and suppression of protein catabolism, favoring better performance during exercise. On the other hand, in the luteal phase, high concentrations of progesterone are found, which has a physiological role during the menstrual cycle, exerting pituitary effects; in addition, the literature suggests that its high concentration affects the fertilization process [4]. When it comes to the role of progesterone in oxidative metabolism, the data are still unclear, suggesting that it behaves antagonistically, inhibiting the metabolic benefits of estrogen, becoming responsible for increased protein catabolism and higher rates of oxidation of some amino acids (AA), such as lysine, alanine, and glutamine [5,6,7]. Furthermore, metabolic flexibility may be associated with variations in concentrations observed during the menstrual cycle, which may mainly influence responses to high-intensity exercises [1,2,3,8,9,10].

The literature has consistently shown that regular cardiorespiratory endurance training improves the performance of tasks that fundamentally depend on oxidative energy metabolism due to the increased ability of skeletal muscles to transport and utilize oxygen (O_2_) and lipids (LIP) when exercised. For this reason, different cardiorespiratory exercise (ECr) protocols have been proposed in order to optimize the use of fatty acids (FAs) since low rates of LIPox may be involved with increased weight gain, obesity, and type II diabetes [9,11]. Different audiences have used high-intensity interval training (HIT) as an alternative method to traditional cardiorespiratory exercise or moderate-intensity continuous training. HIT consists of high-intensity bursts interspersed with periods of passive recovery or low intensity [12,13], which have shown similar and even more efficient results in cardio-respiratory and metabolic parameters in a shorter intervention time [8,13].

When analyzing the literature, it becomes noticeable that few studies with HIT or other training methods associate women’s energy metabolism in different phases of the menstrual cycle with possible modulation of metabolic flexibility and different impacts on health. Nevertheless, this combination is critical in preventing cardiovascular diseases [14] and improving physical fitness [14,15,16]. Thus, the present study aimed to analyze the metabolic flexibility of women of reproductive age and to compare the influence of HIT on the rates of oxidation of energy substrates in different phases of the menstrual cycle.

## 2. Materials and Methods

### 2.1. Trial Design, Setting and Ethics

This is a pre-post pilot study on the effects of the same intervention (exercise) in two groups (women at different stages of the menstrual cycle, called follicular and luteal; FL and LT, respectively), comparing pre-training with post-training variables. The study was not randomized and did not have a placebo or control group. The study was carried out at the Physical Education Research Laboratory of the University City of São Paulo, between August 2018 and December 2019, after approval by the Research Ethics Committee (CAEE: 31998914.8.0000.0064). All participants signed an informed consent form for inclusion in the study, anthropometric measurements, and intervention (physical training).

### 2.2. Inclusion and Exclusion Criteria

We included in this study all women aged 18–35 years with a body mass index (BMI) of 18.5–24.9 kg/m^2^, eumenorrheic, and were not using oral contraceptives (OCs). The women had to present a minimum physical fitness to be included in this study (active or irregularly active), according to the International Physical Activity Questionnaire (IPAQ) [17]. Smokers, those under drug treatment for weight control, hormonal abnormalities, and who stopped using hormonal contraceptives for less than six months were excluded.

### 2.3. Outcomes Evaluations

After the interview, clinical evaluation, and anthropometric measurements, the women were conveniently assigned to the follicular or luteal phase after entering the data in the Flo application. Then, they were tested to determine peak oxygen consumption (VO2peak), peak velocity (Vpeak), and ventilatory anaerobic thresholds (VATs: VAT1 and VAT2). Subsequently, they were submitted to running exercise at submaximal work pace for 45 min (SWRE) for indirect calorimetry, as described below, both tests in the same phase of the cycle identified initially. Subsequently, the women performed eight training sessions, with 48 h of recovery. At the end of the training period, the women had their data entered again in the Flo application, to identify whether they were in the same phase that they were initially evaluated, if they were, they performed the final tests for new determinations of the VAEs and performed another SWRE, the cases not were in the same phase of the initial exams, they waited until they were in the same phase, to be reassessed. Only the training sessions were not performed only in the corresponding menstrual cycle phase (Figure 1).

Three physical education professionals were trained to carry out the assessments and reassessments. Each of them always performed the same procedure: anamnesis, anthropometric measurements, or body composition. The participants were blinded to the groups they belonged to, and so were the professionals who evaluated their data.

### 2.4. Body Composition

Body composition was determined using a bioimpedance device (Bio-dynamics 310), according to the manufacturer’s instructions, to analyze body fat percentage (%BF), fat mass (FM), fat-free mass (FFM), and basal metabolic rate (BMR). The distance of 5 cm between the electrodes was respected, and the participant was positioned in dorsal decubitus with the right foot and hand slightly away from the trunk [18,19]. Twenty-four hours before the test, participants were not allowed to consume alcoholic beverages or stimulants and had to perform good hydration. In addition, they had to abstain from strenuous physical activity 12 h before the analysis, fast for 2–3 h, and not be on their period. Finally, immediately before the start of the test, the participants emptied their bladders.

### 2.5. Cardiopulmonary Test

The test protocol consisted of running on a treadmill (Model ATL, Inbrasport Ltd., Porto Alegre, Brazil) with an initial speed of 6 km/h followed by increments of 1 km/h every minute until the participants’ voluntary exhaustion. Ventilatory parameters were collected throughout the tests at each respiratory cycle and analyzed at an average of 20 s using computerized gas analyzers (model VO2000; Inbrasport Ltd., Porto Alegre, Brazil). The gas analyzer was calibrated to standard volume and gas concentration immediately before the day’s first test and recalibrated after each one, according to the manufacturer’s standardization. Heart rate (HR) was continuously recorded using a heart monitor (Sport Test model; Polar Electro OY; Kempele, Finland) throughout the tests. The Borg subjective perception of exertion scale was used to help monitor the intensity of the test [20]. After exhaustion, two and a half minutes of recovery were performed, with 25% of the maximum speed reached. During recovery periods, only HR was monitored. The criteria for determining VO2_peak_ and exhaustion were: the occurrence of a plateau in VO_2_ (characterized by an increase of 2 mL/kg/min or less) and the inability to maintain running speed, respectively. Vpeak corresponded to the highest speed reached during the tests. The VATs were determined according to Marquezi et al. [21].

### 2.6. Indirect Calorimetry (SWRE)

The balance of energy substrates was calculated by indirect calorimetry during submaximal work-rate running exercise for 45 min (SWRE), performed after six hours of fasting, before and after the training period. The dietary restriction before the SWRE session aimed to prevent the oxidation of exogenous carbohydrates among the participants during the experimental sessions [22]. Ventilatory parameters were collected along the SWRE at each respiratory cycle and analyzed at an average of 20 s using a computerized gas analyzer (model VO2000; Inbrasport Ltd., Porto Alegre, Brazil) to determine the LIPox and CHOox rates.

LIPox and CHOox rates were determined in 5-min blocks along the SWRE (5 to 40 min) from the VO_2_ and VCO_2_ (L/min) mean values, corresponding to the last 2 min of each block. Oxidation rates in g/min were calculated using stoichiometric equations from Frayn (1983), assuming an insignificant nitrogen excretion rate. The energy from LIPox and CHOox (LIPkc and CHOkc, respectively; in kcal/min) was calculated from their respective energy equivalents (9.75 and 3.87 kcal/g; for LIP and CHO, respectively [23].

### 2.7. Training Sessions

Participants performed eight HIT sessions, with a 48h interval between sessions. The treadmill HIT protocol consisted of two initial warm-up periods of 2 min each at 25% and 50% V_peak_, followed by eight sets of 60 s at 100% V_peak_ for 75 s of passive recovery, plus two cool-down periods of 2 min each, at 50% and 25% of V_peak_. Ventilatory parameters and heart rate (HR) were measured in each training session to determine the relative intensity of effort [21]. The training sessions were held between 12 p.m. and 7 p.m. to meet the availability of the participants.

### 2.8. Statistical Analysis

The results are presented as the mean values ± standard error. The homogeneity and normality of the variables were verified using the Bartlett’s test and Shapiro–Wilk test, respectively. When appropriate, participants’ characteristics, ventilatory parameters, HR, and running speed were compared using Student’s *t*-test or Sing’s test for paired data. The VAT means values relative to peak effort for VO_2_, HR, and VEL pre- and post-training tests between groups were analyzed by one-way ANOVA and Tukey HSD post hoc test.

The substrate oxidation rates of pre- and post-training according to the monthly sexual cycle phases were analyzed using multiple comparisons by Kruskal–Wallis test. The significance level adopted was *p* < 0.05. Statistical treatment was performed using the Statistica for Windows software (version 8.0, 2007; Statsoft, Inc. Tulsa, OK, USA).

## 3. Results

Initially, the anthropometry and body fat percentage (%BF) of 19 women were evaluated, of which 16 had BMI and %BF according to the inclusion criteria. Among these, eight were directed to the follicular group (FL) and eight to the luteal group (LT). Participants were allocated according to their menstrual cycle phase, on the day of the first indirect calorimetry assessment, after analyzing the data entered in the Flo application (1st to 14th day: follicular group/15th to 28th day: group luteal). Later on, two participants from the FG and three from the GL dropped out. The flow of participants is shown in Figure 2.

### 3.1. Characterization of Participants

Anthropometric, body composition, resting blood glucose, and ventilatory data are presented in Table 1. The data are considered normal for the age and sex. No significant differences were found between the groups.

### 3.2. Max Tests and Exercise Intensity

The absolute mean values of peak effort and VATs for VO_2_, HR, and running speed (VEL) of maximal pre- and post-training tests are presented in Table 2. No significant differences were observed between the absolute values in the pre- and post-training peak effort for VO2, HR, and VEL. The VAT2 velocity was 16.98% higher post-training for the FL group.

The VAT means values relative to peak effort for VO_2_, HR, and VEL pre- and post-training tests are presented in Table 3. No significant differences were observed.

### 3.3. Training Sessions

During training sessions, V_peak_ was ~13.5 km/h, which produced relative intensities of ~89% of VO_2peak_ for both groups. There was a significant difference in relative HR intensity between groups (Table 4); the relative HR intensity was 4.36% higher for the FL group.

### 3.4. Oxidation of Substrates

The indirect calorimetry data are presented in Table 5. The comparison showed significant differences in relative energy derived from pre- and post-training between groups of −61.42% and −59.26% from CHO, respectively, and 27.46% and 34.41% from LIP, respectively (Table 6).

The relative energy derived from CHO after the training period was 18.89% higher in the FL group and 25.50% higher in the LT group. Consequently, the relative energy derived from LIP after training was 8.45% and 3.46% lower in FL and LT groups, respectively (Table 6 and Figure 3).

## 4. Discussion

Our purpose was to examine the substrate oxidation rates of reproductive-age eumenorrheic women by indirect calorimetry during SWRE at two hormonally distinctive menstrual cycle phases, pre- and post-HIT. The present substrate metabolism finding indicated a greater dependency upon lipid versus carbohydrate in the menstrual cycle luteal phase compared to the follicular phase in pre-intervention. This finding agrees with other studies that measured exercise RER at the low and moderate intensities of maximal oxygen consumption (VO_2max_).

Hackney et al. reported greater LT lipid oxidation only when exercise was performed at 35–60% of the VO_2max_, but not at the intensity of 75% of the VO_2max_ in healthy eumenorrheic women, post 30-min treadmill run [24]. In another study, Wenz et al. also found greater LT lipid oxidation during cycle ergometer exercise at 30 and 50% of the VO2max, but not at 70% in healthy eumenorrheic women [25]. The literature has shown that the hormone concentrations present in the different menstrual cycle phases significantly influence the energy metabolism rates, with changes in estrogen secretion, which generates inflammatory reactions during and after physical exercise [2,5,7,26].

During FL there is a high concentration of 17β-estradiol, progressive increase of estrogen, and absence of progesterone, which favors the absorption of glucose by the muscle and its oxidation, reducing LIPox and protein oxidation [5,7,26]. On the other hand, in TL, a peak of estrogen and mainly of progesterone generates negative impacts of CHOox, forcing the body to carry out greater oxidation of lipid structures and muscle proteins [2,5,26,27,28].

When the reactions of these hormones to different exercise intensities were studied, variations in substrate oxidation rates were mentioned, in which low rates of CHOox and high rates of LIPox were found in the LT in low and moderate-intensity exercises [2,8,29]. On the other hand, in high-intensity exercises, in which there is a greater energy demand, there is a high production of endogenous glucose and a significant effect of estrogen on hepatic glucose production [8]. In addition, a high concentration of 17β-estradiol during these exercises generates an increase in muscle glycogen storage and oxidation capacity, which is considered one of the main mediators in the preference for using substrates to improve performance during exercise, mainly influencing the use of carbohydrates in women [2,5,8].

In contrast, other studies have found greater carbohydrate oxidation and lower lipid oxidation in LT compared to FL during the menstrual cycle with exercise intensity of 45% [30], 50% [31], and 65% [30,32] of the VO2max in physically active and healthy women [29]. This finding suggests some inability to switch between oxidative and glycolytic metabolism, described as metabolic inflexibility, a phenomenon not specific to adults or associated only with insulin resistance, but also linked to the high concentration of progesterone in this phase of the cycle, while estrogen promotes sensitivity to insulin and possibly increases glycogen storage [5,33,34,35,36]. According to Aparecido et al. [26], insulin-resistant individuals are initially normoglycemic.

For Vaiksaar et al. [36], the reasons for these inconsistencies related to the menstrual cycle phase effects on exercise fuel oxidation could be the discrepancies in the exercise mode and protocol, the RER measurements during the exercise, the pretrial diet, and exercise control. However, in the present study, RER measurements were controlled throughout the protocol, and the pre-calorimetry-fed state of the participants was equalized after dietary restriction with maltodextrin intake.

The present study’s results show that eight HIT sessions promoted beneficial effects for other parameters, such as Vpeak in FL. Data that corroborate those presented by Gibala et al. [35], who, after a short HIT program, observed an improvement in cardiorespiratory fitness associated with peripheral adaptations, such as increased mitochondrial biogenesis and improved muscle buffering capacity.

According to Aparecido et al. [26], changes in central factors, such as maximum cardiac output, total hemoglobin, and plasma volume, are affected by long-term HIT (>12 weeks) because longer interventions are needed to induce adaptations. It explains the absence of significant differences in VO2peak and VAT intensities in the present study between the groups after only eight training sessions.

The comparison between groups showed significant differences in relative energy derived from CHOox pre- and post-training of −61.4% and −59.3%, respectively, and LIPox pre- and post-training of 27.5% and 34.4%, respectively. The relative energy derived from CHOox after the training period was 18.9% and 25.5% higher for FPG and LPG, respectively; consequently, the relative energy derived from LIPox after the training period was 8,45% and 3.46% lower for FL and LT, respectively. The greater sensitivity of alpha- and beta-adrenergic receptors, with increased glucose uptake, greater glycolytic enzyme activation, and modulation of substrate oxidation due to moderate-to-high strength or resistance exercise can explain this phenomenon [37]. According to Jeppesen and Kiens [38], these adaptations result from greater recruitment of type II fibers and blood adrenaline concentrations, which stimulate glycogen degradation, glycolysis, and consequently, lactic acid production.

This discrepancy in the results of the studies reinforces that the impact of HIT on women of reproductive age is still controversial. This incongruity can be justified by the wide variety of HIT protocols applied or even by the inherent characteristics of the female sex—totally ignored in the methodology and discussion of the studies selected [26]. Currently, there is evidence that changes in ovarian hormones (estrogen and progesterone) can influence several physiological/metabolic factors related to physical exercise, especially in high intensity and short duration (as, for example, seems to occur in lipoprotein metabolism, glucose metabolism and the modulation of inflammatory responses after physical training) [26].

Studies have demonstrated that estrogen has a direct and indirect (via growth hormone [GH] and cortisol) lipolytic action on adipose tissue. Several studies have reported elevated resting and exercise levels of GH in women during the LT. However, it was recently reported that there is no enhanced GH response to exercise in the LT compared to the FL. Data from the same study also showed increased cortisol during exercise in the LT but no increase during the FL [24]. Unfortunately, we were unable to measure GH, cortisol, and sex hormones from our blood samples, so we cannot shed further light on this issue.

At this point, it seems important to investigate women’s hormonal status and physiological exercise-induced responses to clarify specific hormonal and metabolic responses leading to specific fuel selection. Moreover, cellular studies must consider the complex inherent mechanisms of such adaptations.

The novelty of the current study was that the oxidation substrate post-HIT ratio and the female characteristics had not been previously investigated and/or discussed in women of reproductive age. Thus, further research is needed to test the potential superiority of HIT vs. moderate-intensity continuous training (MICT) on body composition and general oxidative metabolism. Moreover, it will be vital to elucidate whether hormonal changes in the female ovarian cycle or the use of different types of contraceptives (estrogen and progesterone) could induce a state of both metabolic “flexibility or inflexibility” effects in this population under different protocols of HIT training.

Our current data, although showing extreme values, probably due to the small sample size, suggest an important relationship between the training model adopted and the adaptations promoted on the oxidation capacity of CHO and LIP throughout the monthly sexual cycle. Undoubtedly, this is an important limitation of this study, which is why we classify it as a preliminary or pilot study.

## 5. Conclusions

The phases of the menstrual cycle promote significant changes in substrate oxidation rates, leading to a decrease in CHOox and an increase in LIPox in the follicular phase, possibly due to the higher concentration of estrogen, which favors changes in metabolism and performance during submaximal work rate exercise. High-intensity interval training can minimize the observed differences and constitute an alternative intervention.

## Figures and Tables

**Figure 1 ijerph-20-03148-f001:**
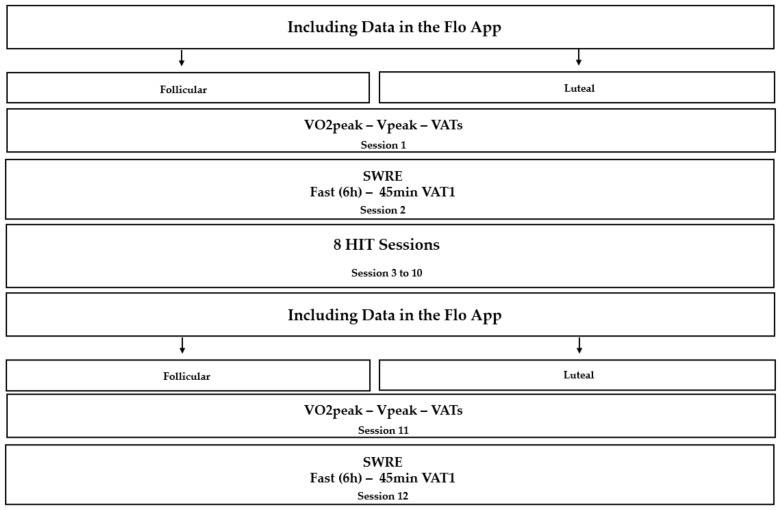
Intervention.

**Figure 2 ijerph-20-03148-f002:**
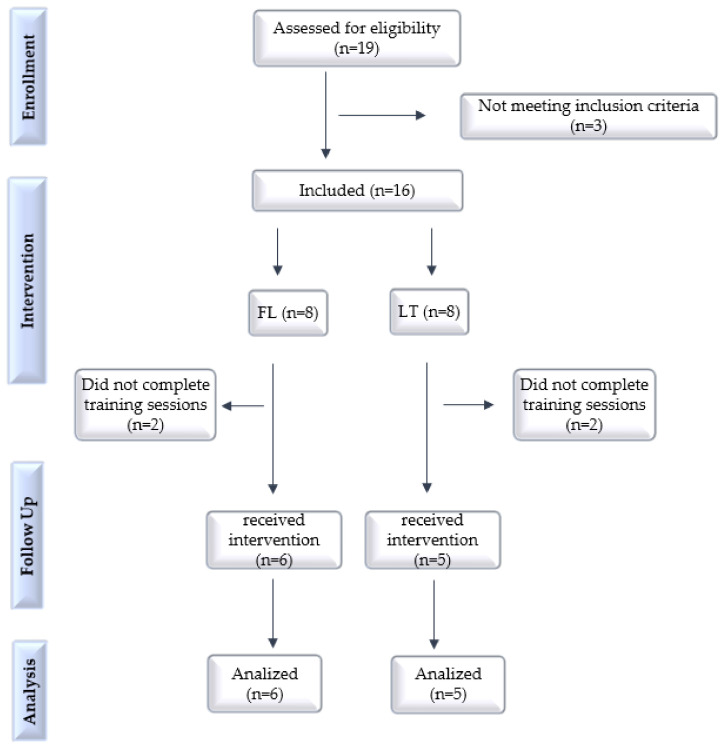
Flowchart of the inclusion, allocation, and follow-up process of the 11 study participants. Follicular (FL) and luteal (LT) groups.

**Figure 3 ijerph-20-03148-f003:**
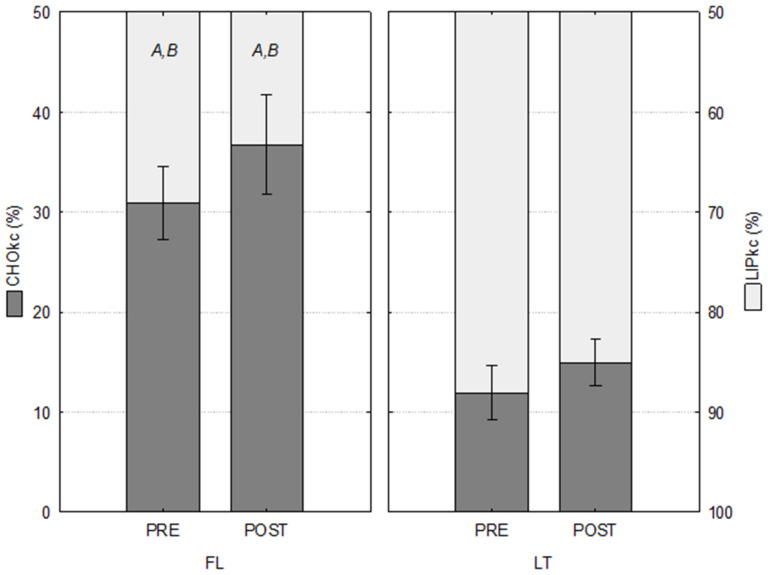
Relative energy derived from carbohydrates (CHOkc) and lipids (LIPkc) pre- and post-training. Mean values ± standard error; follicular (FL, *n* = 6) and luteal (LT, *n* = 5). ^A^ = *p* < 0.05 vs. LT PRE; ^B^ = *p* < 0.05 vs. LT POST.

**Table 1 ijerph-20-03148-t001:** Baseline data of groups follicular (FL) and luteal (LT).

	FL (*n* = 6)	LT (*n* = 5)
Age (years)	22.50 ± 0.81	21.20 ± 1.07
Height (m)	1.60 ± 0.02	1.57 ± 0.03
Body mass (kg)	54.67 ± 3.60	55.04 ± 2.04
Body fat (%)	22.42 ± 2.97	26.24 ± 0.85
Resting blood glucose (mg/dL)	83.05 ± 5.06	87.60 ± 5.64
VO_2_peakPRE (L/min)	1.93 ± 0.17	1.80 ± 0.09
VO_2_peak/kg PRE (mL/kg/min)	34.83 ± 2.42	32.92 ± 1.84
VO_2_peak POST (L/min)	2.16 ± 0.19	1.97 ± 0.07
VO_2_peak/kg POST (mL/kg/min)	39.22 ± 1.45	35.77 ± 1.75
Body mass index (kg/m^2^)	21.36 ± 1.64	22.44 ± 0.63

Mean values ± standard error. No significant differences were found.

**Table 2 ijerph-20-03148-t002:** Absolute mean values of peak effort and VATs.

			VO_2_ (mL/kg/min)	HR (bpm)	VEL (km/h)
FL (*n* = 6)	VAT1	PRE	14.29 ± 1.24	140.00 ± 4.91	6.83 ± 0.17
		POST	17.22 ± 1.23	130.83 ± 7.61	7.00 ± 0.00
	VAT2	PRE	25.65 ± 1.20	164.00 ± 5.20	8.83 ± 0.17 ^A^
		POST	30.38 ± 1.12	169.50 ± 5.31	10.33 ± 0.21
	PEAK	PRE	34.83 ± 2.42	186.33 ± 2.29	12.50 ± 0.34
		POST	39.22 ± 1.45	188.67 ± 3.04	13.83 ± 0.40
LT (*n* = 5)	VAT1	PRE	13.19 ± 0.71	140.20 ± 11.96	7.00 ± 0.32
		POST	14.23 ± 1.94	136.00 ± 7.57	7.00 ± 0.00
	VAT2	PRE	25.13 ± 1.93	169.40 ± 5.97	9.20 ± 0.37
		POST	27.13 ± 1.11	170.40 ± 1.83	10.20 ± 0.37
	PEAK	PRE	32.92 ± 1.84	187.20 ± 4.36	11.80 ± 0.49
		POST	35.74 ± 1.76	187.60 ± 1.50	13.00 ± 0.45

Mean values ± standard error. VAT1: first ventilatory threshold; VAT2: second ventilatory threshold; VO_2_: oxygen consumption; HR: heart rate; VEL: velocity. ^A^ = *p* < 0.05 vs. POST.

**Table 3 ijerph-20-03148-t003:** Ventilatory anaerobic thresholds means values relative to peak effort.

			O2 (%)	HR (%)	VEL (%)
FL (*n* = 6)	VAT1	PRE	47.42 ± 4.30	70.83 ± 3.74	54.94 ± 2.37
		POST	44.7 ± 3.06	69.76 ± 3.56	52.20 ± 1.89
	VAT2	PRE	74.26 ± 2.41	87.98 ± 2.27	72.29 ± 2.13
		POST	77.67 ± 2.12	89.48 ± 1.85	75.58 ± 2.38
LT (*n* = 5)	VAT1	PRE	40.49 ± 2.93	74.52 ± 5.03	59.44 ± 2.18
		POST	39.98 ± 5.29	72.5 ± 3.99	54.10 ± 1.86
	VAT2	PRE	76.02 ± 2.05	90.44 ± 1.98	78.04 ± 1.72
		POST	76.18 ± 2.29	90.88 ± 1.60	78.52 ± 1.38

Mean values ± standard error. VAT1: first ventilatory threshold; VAT2: second ventilatory threshold; O2: oxygen consumption; HR: heart rate; VEL: velocity. No significant differences were found.

**Table 4 ijerph-20-03148-t004:** Mean relative intensities of training sessions.

	O2 (%)	HR (%)
FL (*n* = 6)	89.17 ± 1.82	95.59 ± 0.83 ^A^
LT (*n* = 5)	89.84 ± 1.93	91.60 ± 1.02

Mean values ± standard error. O2, oxygen consumption; HR, heart rate. ^A^ = *p* < 0.05 vs. LT.

**Table 5 ijerph-20-03148-t005:** Indirect calorimetry data.

		TTkc (kcal)	CHOkc (%)	LIPkc (%)
FL (*n* = 6)	PRE	4.25 ± 0.29	30.89 ± 3.62 ^A,B^	69.11 ± 3.62 ^A,B^
	POST	4.54 ± 0.28	36.73 ± 5.01 ^A,B^	63.27 ± 5.01 ^A,B^
LT (*n* = 5)	PRE	4.05 ± 0.23	11.92 ± 2.72	88.08 ± 2.72
	POST	3.58 ± 0.09	14.96 ± 2.30	85.04 ± 2.30

Mean values ± standard error. TTkc: total caloric expenditure; CHOkc: relative amount of energy derived from CHO oxidized; LIPkc: relative amount of energy derived from LIP oxidized. ^A^ = *p* < 0.05 vs. LT PRE; ^B^ = *p* < 0.05 vs. LT POST.

**Table 6 ijerph-20-03148-t006:** Relative substrate oxidation differences between training periods and menstrual cycle phases.

		CHOkc (%)	LIPkc (%)
FL × LT	PRE	−61.42 ^A^	27.46 ^A^
	POST	−59.26	34.41
PRE × POST	FL	18.89 ^B^	−8.45 ^B^
	LT	25.50	−3.46

Mean values ± standard error. CHOkc: relative amount of energy derived from CHO oxidized; LIPkc: relative amount of energy derived from LIP oxidized. ^A^ = *p* < 0.05 vs. POST; ^B^ = *p* < 0.05 vs. LT.

## Data Availability

The data presented in this study are available on request from the corresponding author. The data are not publicly available due to privacy.

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
