# Peer review of "Effect of Menstrual Cycle Phase on Fuel Oxidation Post HIT in Women Reproductive Age: A Pilot Study"

_ijerph, 2023, doi:10.3390/ijerph20043148_

Round 1

Reviewer 1 Report

The paper would have been of great interest in a metabolic or sports medicine journal. However, it has several fundamental flaws:
The introduction lacks specific data on the effect of progesterone on basal metabolism.
It is unfortunate that the study was not randomized, as the authors themselves admit.
It would need to be clarified whether the initial study of the participants was performed during the same phase of the cycle, since weight and body composition change during the cycle.
Table 1. shows that participants classified as LT had significantly higher body fat mass than FT (although it is written that there were no statistical differences!), which could have influenced the results.
From Figure 1. and the study description, it is clear that the entire cycle of the testing in each participant was almost the entire length of the menstrual cycle. Thus, it must have included both the follicular and luteal phases.
The study is based on too few female participants, which does not allow statistically significant conclusions to be drawn. Thus, the study can only be treated as preliminary or pilot (of course, after taking into account the time of exercise limited to one particular phase of the cycle!).
The conclusions should say which phase of the menstrual cycle causes the lowering of CHOox.

Authors' responses to reviewer 1:

The paper would have been of great interest in a metabolic or sports medicine journal. However, it has several fundamental flaws:

The introduction lacks specific data on the effect of progesterone on basal metabolism.

On the other hand, in the luteal phase, high concentrations of progesterone are found, which has a physiological role during the menstrual cycle, exerting pituitary effects; in addition, the literature suggests that its high concentration affects the fertilization process [4]. When it comes to the role of progesterone in oxidative metabolism, the data are still unclear, suggesting that it behaves antagonistically, inhibiting the metabolic benefits of estrogen, becoming responsible for increased protein catabolism and higher rates of oxidation of some amino acids (AA), such as lysine, alanine, and glutamine [5-7]. Furthermore, metabolic flexibility may be associated with variations in concentrations observed during the menstrual cycle, which may mainly influence responses to high-intensity exercises [1-3,8-10].

It is unfortunate that the study was not randomized, as the authors themselves admit.

The participants were allocated among the experimental groups by convenience, that is, according to the phase of the monthly sexual cycle that they presented at the time of the clinical evaluation and anthropometric measurements.

It would need to be clarified whether the initial study of the participants was performed during the same phase of the cycle, since weight and body composition change during the cycle. From Figure 1. and the study description, it is clear that the entire cycle of the testing in each participant was almost the entire length of the menstrual cycle. Thus, it must have included both the follicular and luteal phases.

After the interview, clinical evaluation and anthropometric measurements, the women were conveniently assigned to the follicular or luteal phase after entering the data in the Flo application. Then, they were tested to determine peak oxygen consumption (VO2peak), peak velocity (Vpeak) and ventilatory anaerobic thresholds (VATs: VAT1 and VAT2). Subsequently, they were submitted to running exercise at submaximal work pace for 45min (SWRE) for indirect calorimetry, as described below, both tests in the same phase of the cycle identified initially. Subsequently, the women performed eight training sessions, with 48 hours of recovery. At the end of the training period, the women had their data entered again in the Flo application, to identify whether they were in the same phase that they were initially evaluated, if they were, they performed the final tests for new determinations of the VAEs and performed another SWRE, the cases not were in the same phase of the initial exams, they waited until they were in the same phase, to be reassessed. Only the training sessions were not performed only in the corresponding menstrual cycle phase (Figure 1).

Table 1. shows that participants classified as LT had significantly higher body fat mass than FT (although it is written that there were no statistical differences!), which could have influenced the results.

%GORDURA = relative body fat mass; 0.28 represents no statistical difference.

The study is based on too few female participants, which does not allow statistically significant conclusions to be drawn. Thus, the study can only be treated as preliminary or pilot (of course, after taking into account the time of exercise limited to one particular phase of the cycle!).

Effect of menstrual cycle phase on fuel oxidation post HIT in women reproductive age: A Pilot Study

The conclusions should say which phase of the menstrual cycle causes the lowering of CHOox.

The phases of the menstrual cycle promote significant changes in substrate oxidation rates, leading to a decrease in CHOox and increase in LIPox in the follicular phase, possibly due to the higher concentration of estrogen, which favors changes in metabolism and performance during submaximal work rate exercise.

Reviewer 2 Report

1. Motivation Section should be added.

2. Abbreviation Section should be reported. 

3. The homogeneity of the variables was verified using the Levene test. Please use Bartlett's test and compare the results.

4. The normality property should be tested.

5. Why the authors used the Completely randomized design (CRD) instead of ANOVA test?.  Explain.

6. Non-parametric plots should be displayed to describe the behavior of data.  

7. Did the real data have extreme or outliers' observations?. Explain.

8. Conclusion Section should be expanded.

Authors' response to reviewer 2:

  1. Motivation Section should be added.

When analyzing the literature, it becomes noticeable that few studies with HIT or other training methods associate women’s energy metabolism in different phases of the menstrual cycle with possible modulation of metabolic flexibility and different impacts on health. Nevertheless, this combination is critical in preventing cardiovascular diseases [13] and improving physical fitness [13-15].

  1. Abbreviation Section should be reported. 

Abbreviation Section

CHO: Carbohydrates

LIP: Lipids

AA: Amino acids

CHOox: Carbohydrate oxidation rates

LIPox: Lipid oxidation rates

CHOkc: Relative amount of energy derived from carbohydrate oxidized

LIPkc: Relative amount of energy derived from lipid oxidized

TTkc: Total caloric expenditure

O2: Oxygen

VO2max: Maximal oxygen consumption

VO2peak: Peak oxygen consumption

VATs: Ventilatory anaerobic thresholds

VAT1: First ventilatory threshold

VAT2: Second ventilatory threshold

HIT: High-intensity interval training

MICT: Moderate intensity continuous training

Vpeak: Peak velocity

HR: Heart rate

ECs: Cardiorespiratory exercise

IPAQ: International physical activity questionnaire

SWRE:  Submaximal work rate running exercise (indirect calorimetry)

FAS: Fatty acids

BMI: Body mass index

%BF: Body fat percentage

FM: Fat mass

FFM: Fat free mass

BMR: Basal metabolic rate

VE: Ventilatory equivalents

FE: Expired fractions

RQ: Respiratory quotient

FL: Follicular phase group

LT: Luteal phase group

GH: Growth hormone

LH: Luteinizing hormone

FSH: Follicle stimulating hormone

OCs: Oral contraceptives

  1. The homogeneity of the variables was verified using the Levene test. Please use Bartlett's test and compare the results.

%GORDURA = relative body fat mass; 0.28 represents no statistical difference.

  1. The normality property should be tested.
  2. Why the authors used the Completely randomized design (CRD) instead of ANOVA test?.  Explain.

The results are presented as mean values ± standard error. The homogeneity and normality of the variables were verified using the Bartlett's test and Shapiro-Wilk test, respectively. When appropriate, participants’ characteristics, ventilatory parameters, HR, and running speed were compared using Student's t-test or Sing's test for paired data. The VAT means values relative to peak effort for VO2, HR, and VEL pre- and post-training tests between groups were analyzed by one-way ANOVA and Tukey HSD post-hoc test. The substrate oxidation rates of pre- and post-training according to the monthly sexual cycle phases were analyzed using multiple comparisons by Kruskal-Wallis test. The significance level adopted was p<0.05. Statistical treatment was performed using the Statistica for Windows software (version 8.0, 2007; Statsoft, Inc. United States).

  1. Non-parametric plots should be displayed to describe the behavior of data.  

Despite the reviewer's suggestion, we found it more appropriate to use the graph below (present in the first manuscript submission) to visualize the substrate oxidation data.

  1. Did the real data have extreme or outliers' observations?. Explain.

Our current data, although showing extreme values, probably due to the small sample size, suggest an important relationship between the training model adopted and the adaptations promoted on the oxidation capacity of CHO and LIP throughout the monthly sexual cycle. Undoubtedly, this is an important limitation of this study, which is why we classify it as a preliminary or pilot study.

  1. Conclusion Section should be expanded.

The phases of the menstrual cycle promote significant changes in substrate oxidation rates, leading to a decrease in CHOox and increase in LIPox in the follicular phase, possibly due to the higher concentration of estrogen, which favors changes in metabolism and performance during submaximal work rate exercise. High-intensity interval training can minimize the observed differences and constitute an alternative intervention.